# The P2X7 Receptor in Autoimmunity

**DOI:** 10.3390/ijms241814116

**Published:** 2023-09-14

**Authors:** Fabio Grassi, Gaia Salina

**Affiliations:** Institute for Research in Biomedicine, Faculty of Biomedical Sciences, Università della Svizzera Italiana, 6500 Bellinzona, Switzerland; gaia.salina@irb.usi.ch

**Keywords:** purinergic signaling, autoimmunity, autoantibodies, T cells

## Abstract

The P2X7 receptor (P2X7R) is an ATP-gated nonselective cationic channel that, upon intense stimulation, can progress to the opening of a pore permeable to molecules up to 900 Da. Apart from its broad expression in cells of the innate and adaptive immune systems, it is expressed in multiple cell types in different tissues. The dual gating property of P2X7R is instrumental in determining cellular responses, which depend on the expression level of the receptor, timing of stimulation, and microenvironmental cues, thus often complicating the interpretation of experimental data in comprehensive settings. Here we review the existing literature on P2X7R activity in autoimmunity, pinpointing the different functions in cells involved in the immunopathological processes that can make it difficult to model as a druggable target.

## 1. Introduction

Autoimmune diseases are chronic disorders characterized by inadequate organismal self-tolerance by the adaptive immune system and tissue-destroying inflammation, thereby reducing life quality and expectancy. The etiology of these conditions is multifactorial, and many elements ultimately contribute to the pathological phenotype. The exact mechanisms underlying most autoimmune conditions are not fully understood, but dysregulation of immune responses and various genetic and environmental factors play a role in their development. Despite multiple efforts directed towards finding effective therapies, research to date has not been able to provide a definitive treatment for the majority of them. Purinergic signaling is an evolutionary conserved mechanism involved in a variety of physiological and pathophysiological processes. Its functions have been intensely studied in the context of immunity and immunopathology, given the widespread expression of purinergic receptors in all types of immune cells. Purinergic receptors are divided into two subtypes based on their binding affinity to different purine derivatives. P1 receptors (A1, A2A, A2B, and A3) are G protein-coupled receptors specific for adenosine. Conversely, P2 receptors are divided into G protein-coupled metabotropic P2Y receptors (P2Y1, 2, 4, 6, 11–14) with binding affinities for ATP, ADP, UTP, UDP, and UDP-glucose and P2X ionotropic ligand-gated channels (P2X1-7) that bind ATP [1]. Among the members of the P2X family, the P2X7 receptor (P2X7R) is one of the most studied and stands out for its relevance in regulating inflammation and immune responses [2]. P2X7R dysfunction has been implicated in the development or progression of several autoimmune conditions. How P2X7R dysregulation could contribute to these diseases, however, is not fully understood. Nevertheless, targeting P2X7R has emerged as a potential therapeutic strategy in some autoimmune and inflammatory conditions.

## 2. The P2X7 Receptor

The P2X7R is a homotrimer with low affinity for ATP, and several lines of evidence suggest it might contribute to the pathogenesis of different autoimmune disorders [3,4]. Its broad expression on immune cells and pleiotropic role in various processes render it often difficult to assign a univocal contribution of P2X7R to immunopathological outcomes. In humans, the P2X7R is encoded by the gene *P2RX7*, which is located on chromosome 12, in close proximity to the gene encoding for the homologous P2X4 receptor, suggesting a possible origin by duplication [5]. *P2RX7* is highly polymorphic; it presents 11 splice variants and over 30 single nucleotide polymorphisms (SNPs) that might affect its activity [6]. Some variants are functional, while others are not; the activity of gain-of-function or loss-of-function variants of the receptor could significantly contribute to the development of pathophysiological phenotypes. The wildtype gene encodes a protein of 595 amino acids, formed by two transmembrane domains (TM1 and TM2) connected by an extracellular loop, an intracellular C-terminus, and an intracellular N-terminus [7]. Three identical subunits are assembled to constitute the active form of the receptor, which is generally triggered by the binding of three molecules of extracellular ATP (eATP) [8]. The P2X7R is characterized by dual gating: exposure to micromolar concentrations of eATP allows this receptor to function as a non-selective cation channel for Na^+^, K^+^, or Ca^2+^, whereas sustained activation with higher concentrations of ATP leads to the formation of a pore permeable to molecules up to 900 Da [9]. The arrangement of this non-selective macropore takes place thanks to the presence of an extended carboxyl-terminal tail of 293 amino acids, which is distinctive of P2X7R and whose absence results in impaired pore formation [10]. Opening of the P2X7R cytolytic pore allows significant transmembrane fluxes of both ions and small molecules that affect cell homeostasis and viability [11]. On the contrary, P2X7R signaling as a cation channel is associated with the activation of several downstream pathways and transcriptional regulation [12]. To date, the mechanism mediating channel-to-pore transition has not been determined, and a number of phenomena have been hypothesized to contribute to the P2X7R-mediated permeability of cells to small molecules [13]. Given the crucial role of P2X7R in governing inflammatory and immunological responses, its activity is tuned by the plasma membrane ectonucleoside triphosphate diphosphohydrolase-1 CD39, which catalyzes the hydrolysis of γ- and β-phosphate residues of ATP and ADP to the monophosphonucleoside derivative AMP, thereby limiting P2X7R stimulation by eATP [14]. Moreover, P2X7R cleavage by matrix metalloproteinase 2 (MMP-2) released by the sustained activity of the receptor halts signaling and pore formation, constituting an additional regulatory layer in the physiology and pathophysiology of P2X7R [15].

### 2.1. P2X7R in Innate Immunity

In cells of the innate immune system, stimulation of P2X7R triggered by eATP released upon activation of pattern recognition receptors (PRRs) by pathogen-associated molecular patterns (PAMPs) or damage-associated molecular patterns (DAMPs) promotes the arousal of several proinflammatory pathways that provide the first line of defense against pathogens [16,17]. The influx of Ca^2+^ and efflux of K^+^ ions induced by channel opening modifies the intracellular microenvironment, triggering nucleotide-binding domain, leucine-rich-containing family, and pyrin domain-containing 3 (NLRP3) inflammasome assembly and activation [6,18]. The triggering of this pathway has been hypothesized to be mediated also by the release of reactive oxygen species (ROS), cathepsins, or by the direct interaction of intracellular domains of the receptor with NLRP3 components [19,20,21]. The inflammasome is a multi-protein complex responsible for the recruitment and proteolytic cleavage of pro-caspase-1 and its transition to active caspase-1 [22,23]. The initiation of this pathway promotes the processing of two pro-inflammatory mediators, IL-1β and IL-18, which are activated and released in the extracellular environment [22,24,25]. The former cytokine is able to bind its receptor on the surface of target cells, activating NF-kB and inducing the expression of genes sustaining inflammation, while the latter promotes the production of IFN-γ and induces type I inflammatory responses [26]. Indeed, murine models lacking P2X7R are characterized by impaired IL-1β release [27]. P2X7R further shapes innate immune system activity, exerting different effects depending on the cell type. For instance, in monocytes and macrophages, P2X7R triggers IL-6 upregulation in a Ca^2+^-dependent fashion [28,29]. P2X7R stimulation is also related to the activation of the MAP kinase pathway, phospholipase D, and the production of reactive oxygen and nitrogen species [29]. Furthermore, P2X7R activation induces dendritic cell (DC) maturation by switching on the NF-kB pathway. The downstream cascade triggered by the transcription factor induces both morphological changes and the release of different inflammatory cytokines, including IFN-γ and IL-12 [30].

### 2.2. P2X7R in Adaptive Immunity

P2X7R is expressed on both T and B lymphocytes. Functions and activities of these two immune cell populations are strictly intertwined since activation of T cell immune response is critical to both induced cellular and humoral immunity. The activation of P2X7R in B cells is generally associated with an active phenotype, pro-inflammatory effects, and IgM production [31]. Moreover, its stimulation induces the activation of matrix metalloproteinases (MMP) responsible for the proteolytic cleavage of the plasma membrane proteins CD62L, CD21, and CD23 [32,33]. CD62L, also known as L-selectin, is a C-type lectin involved in adhesion to endothelial cells and transendothelial migration, promoting B cell recruitment into secondary lymphoid organs. CD23 is a receptor for immunoglobulin E (IgE) entangled in the development and functions of B cells [34]. CD21 is complement receptor 2 (Cr2), which affects BCR-mediated signaling [35]. Therefore, research to date suggests a role for P2X7R in the modulation of B cell migratory behavior, activation, and antibody production. 

P2X7R is expressed in both CD4^+^ and CD8^+^ T cells and represents an important regulator of T cell development, functions, and memory formation [36,37]. During adaptive immune responses, T cells are activated by the binding of the T cell receptor (TCR) to cognate antigen exposed within the major histocompatibility complex (MHC) expressed by antigen-presenting cells (APCs), and this interaction is adjuvated by other costimulatory signals that condition the outcome of T cell activation [38,39]. Stimulation of naïve CD4^+^ T cells results in the release of ATP via pannexin-1 hemichannels, resulting in autocrine and paracrine triggering of P2X7R (and other P2XR subtypes). The activated downstream pathway leads to Ca^2+^ influx, nuclear factor of activated T cells (NFAT) activation, and mitogen-activated protein kinases (MAPK) signaling, promoting IL-2 expression and T cell proliferation [40,41]. Moreover, P2X7R activation is associated with the shedding of CD62L, CD27, and IL-6R [42]. CD62L fulfills the same functions in both B and T cells (see above), while CD27 is a type I transmembrane protein of the tumor necrosis factor (TNF) receptor family acting as a costimulatory receptor, supporting T cell function and memory formation [43,44]. IL-6R removal could condition T cell polarization and function. Analogously to B cells, the shedding of these membrane proteins is mediated by the activity of matrix metalloproteases (MMPs) [45]. T regulatory (Treg) cells play a pivotal role in maintaining tolerance to self-antigens; they express high surface levels of P2X7R, and autocrine or paracrine activation by ATP is able to inhibit their suppressive potential, probably by limiting *Foxp3* expression [46]. In particular, inhibition of P2X7R or deletion of the receptor itself leads to preservation of Treg’s functional profile and amelioration of inflammation in chronically inflamed tissues. P2X7R activity undermines Treg cells stability, induces their conversion to T helper 17 (Th17) cells [46], and plays an important role in eATP-mediated apoptosis of this T cell subset [47]. Moreover, the expression and activation of P2X7R on cells from the innate compartment promote IL-6 and IL-1β secretion, which favor the mounting of the Th17 response [48]. T cell-dependent humoral immunity is also robustly influenced by P2X7R activity; deletion of the *P2rx7* gene in mice leads to reduced cell death of T follicular helper (Tfh) cells, resulting in increased germinal center (GC) formation in gut-associated lymphoid tissue (GALT) and deregulated production of secretory IgA (sIgA). Conversely, in wildtype mice, P2X7R in GALT Tfh cells is downregulated by acute TCR signaling, allowing adaptation of the SIgA response to emerging bacterial species and favoring the establishment of a beneficial, diverse microbiota [49,50]. An analogous regulation of P2X7R expression has been shown to operate in tissue resident memory T (Trm) cells, suggesting regulated P2X7R expression could represent a general strategy for focusing the effector T cell response to possibly dangerous invaders while avoiding energy consumption by irrelevant T cells via P2X7R-mediated cell death induction [51]. On the other hand, impaired P2X7R activity could lead to overactivation of effector T cells with beneficial and detrimental effects in cancer and autoimmunity, respectively [52,53].

## 3. The P2X7R in Autoimmunity

Autoimmunity results from the failure of mechanisms that ensure tolerance of self-antigens, resulting in the generation of adaptive immune responses targeting healthy tissues. Self-tolerance can be divided into two broad etiological categories referred to as central and peripheral tolerance. The former takes place in primary lymphoid organs, namely bone marrow and thymus, where discrimination for clonal reactivity to self-antigens of differentiating lymphocytes and elimination of autoreactive cells are generally controlled by the affinity of the antigen receptor. In this respect, medullary thymic epithelial cells (MTECs) are important elements regulating central tolerance by ectopically expressing peripheral antigens under the control of the autoimmune regulator (AIRE) protein, a transcriptional factor promoting the expression of selected tissue-specific antigens (TSAs). This characteristic of MTECs results in the exposure of differentiating T cells to a wide repertoire of self-tissue antigens and the elimination of thymocytes reacting with high affinity to autoantigens by inducing cell death [54]. Conversely, peripheral tolerance occurs in tissues to prevent the activation of lymphocytes that escape central tolerance and exploits several mechanisms, among which active suppression, the presence of inhibitory coreceptors, and the induction of anergy or ignorance induced by antigen sequestration [55]. In this respect, it is worth mentioning that AIRE is able to promote the differentiation of self-reactive thymocytes into immunosuppressive natural Treg cells expressing CD4 and CD25 [56]. The relevance of AIRE in regulating adaptive immune tolerance is underscored by the evidence that AIRE mutations contribute to the onset of a wide spectrum of organ-specific autoimmune diseases, among which are systemic sclerosis, rheumatoid arthritis (RA), and others [57].

Given its pleiotropic function in shaping the activity of the immune system, P2X7R has been implicated in the pathogenesis and progression of autoimmunity; however, its contribution remains controversial since both beneficial and detrimental effects have been observed. Certainly, its role in inflammation and innate immunity is relevant in many autoimmune diseases, as P2X7R-mediated secretion of proinflammatory cytokines, chemokines, and other soluble mediators can fuel the autoimmune reaction [58]. Therefore, exploiting P2X7R as a potential therapeutic target could represent a promising opportunity to ameliorate the pathological manifestations suffered by many patients affected by inflammatory and autoimmune disorders.

### 3.1. Systemic Lupus Erythematosus (SLE)

SLE is a systemic and multifactorial autoimmune disorder characterized by the presence of autoantibodies directed against nuclear components. The binding of antinuclear antibodies (ANAs) to their antigens leads to the formation of immune complexes that are deposited within tissues, mediating the multiple organ damage observed in SLE patients [59]. In this context, P2X7R might play a dual function: it is responsible for the triggering of the NLRP3 inflammasome cascade and is pivotal in inducing cell death, which sustains the inflammatory loop and the release of autoantigens [60]. In fact, the leakage of nuclear components targeted by autoantibodies is mediated by the dysregulation of different mechanisms responsible for cell death [3]. Moreover, the involvement of almost all the components of the inflammasome pathway in SLE pathogenesis has been suggested in a number of experimental approaches provided in both animal models and human patients [3].

Several induced and spontaneous murine models have been developed to experimentally reproduce SLE pathogenesis in vivo [61]. They are generally defined by the presence of autoantibodies, immunocomplex-mediated glomerulonephritis, and eventually lymphadenopathy; essentially, they provided evidence of the association between SLE manifestations and alterations in P2X7R and the inflammasome pathway. Indeed, MLR/*lpr* mice are characterized by increased expression of P2X7R, NLRP3, and ASC in renal tissue. In addition, P2X7R inhibition or deletion significantly reduced the titer of circulating ANAs, attenuated glomerulonephritis severity, and favored the decrease of IL-1β and IL-18 levels and the Th17/Treg ratio [62,63]. The relevance of inflammasome activation in SLE pathogenesis was corroborated in the pristane-induced model, where caspase-1 deletion reduced autoantibody levels and inflammation [64]. Mice with deletions of the gene encoding IL-1β injected with anti-dsDNA antibodies were also resistant to SLE development [65]. Among the multiple genetic loci that have been suggested to contribute to SLE susceptibility, the locus 12q24, which harbors the *P2RX7* gene, has been proposed to be involved in the propensity to develop the disease in Hispanic and European American families [66], while some *P2RX7* polymorphisms were identified as possible inducers of SLE susceptibility in the Chinese population [67] (Table 1).

Different studies revealed upregulation of P2X7R in renal and tubular cells obtained from patients’ biopsies [76], while increased concentrations of IL-1β and IL-18 in the sera of SLE patients correlated with disease severity [77,78]. Adaptive immunity was hypothesized to be involved in the detection of increased levels of IL-17-producing cells in peripheral blood from SLE patients [79]. In line with this, the overexpression of P2X7R on Th1 and Th17 cells has been associated with an exacerbation of the disease [80]. In contrast with the previous hypothesis, Mellouk and colleagues have highlighted the crucial role of P2X7R in regulating T cell homeostasis and demonstrated its involvement in preventing SLE onset by limiting lymphoaccumulation [81]. In fact, P2X7 stimulation promotes caspase-mediated pyroptosis of Tfh cells and controls the development of pathogenic ICOS^+^ IFN-γ–secreting cells. Furthermore, circulating Tfh cells from patients with SLE but not primary anti-phospholipid syndrome patients, a nonlupus systemic autoimmune disease, were hyporesponsive to P2X7R stimulation and resistant to P2X7R-mediated inhibition of cytokine-driven expansion. These results suggested that P2X7R could limit the progressive amplification of pathogenic autoantibodies and could be considered a checkpoint regulator of Tfh cells [53] (Figure 1).

### 3.2. Rheumatoid Arthritis (RA)

RA is a chronic condition defined by the presence of synovial inflammation and progressive articular damage, eventually leading to invalidating complications such as swelling, stiffness, pain, and deformity. RA can also have consequences for other organs, mainly the skin and lungs [82]. Patients are characterized by circulating antibodies targeting citrullinated peptides (anti-CCP antibodies, or ACPAs) [83]. Similarly to other autoimmune disorders, inflammasome activation mediated by P2X7R might play a role in the pathogenesis of this disease [84]. Indeed, the *P2RX7* locus has been associated with susceptibility to developing RA. Notably, some SNPs determining increased activity of P2X7R and its downstream pathway were associated with disease susceptibility and onset [68,73] (Table 1). This hypothesis was corroborated by experiments performed in different murine models of RA. In the rat streptococcal cell wall (SCW) model, the development of arthritis was correlated with high levels of P2X7R in inflamed synovial tissues [85]. In collagen-induced arthritis (CIA), the injection of complete Freund’s adjuvant (CFA) in combination with type II collagen (CII) induced the development of arthritis, and, in this context, P2X7R activation triggered the production of inflammatory cytokines, promoting Th17 differentiation; the use of the P2X7R antagonist A-438079 reduced IL-17 production and alleviated disease severity [86]. Although a specific involvement of P2X7R was not addressed, Ardissone et al. have shown that systemic P2X inhibition by oxidized ATP resulted in a sustained reduction in disease activity, which was associated with a significant decrease in CD3 T cell infiltration in arthritic lesions and a significant amelioration of cartilage erosion. Moreover, circulating autoantibodies directed against mouse CII were significantly reduced, and serum autoantibody levels were significantly correlated with the clinical efficacy of oxidized ATP. These results corroborate the hypothesis that P2X receptor antagonism is endowed with important therapeutic potential in chronic inflammatory rheumatic disorders [87]. Past studies performed on patients affected by RA indicated increased levels of P2X7R-expressing peripheral blood mononuclear cells (PBMCs) in circulation compared to healthy donors [88]. Furthermore, the expression level of the receptor in the synovial tissue could represent a good diagnostic tool since it positively correlates with IL-1β and IL-6 production [4]. Supporting the relevance of P2X7R activity in RA pathogenesis, Li and colleagues demonstrated that, in patients affected by RA, an increased number of Th17 cells expressing the receptor was associated with increased disease activity and the presence of disease biomarkers [80]. A note of caution should be considered when envisaging P2X7R as a target for modulating adaptive immunity in RA. Similarly to SLE, Felix and colleagues demonstrated that *P2RX7* deletion could lead to the unleashing of Tfh cell responses and consequent B cell differentiation and antibody production, thereby enhancing arthritis activity [89] (Figure 1).

P2X7R triggering induces the release of several molecules, contributing to the activation of different pathways that participate in the induction of chronic inflammation and the onset of autoimmunity. For instance, P2X7R-mediated release of transglutaminase 2 (TG2) in the extracellular environment contributes to sustaining inflammation by promoting TGF-β activation [90]. Moreover, TG2 mediates protein alterations, creating immunogenic neo-epitopes that might contribute to the development of RA (Figure 1). This hypothesis was supported by a study performed in the CIA model, which showed that increased levels of TG2 exacerbated the disease [91]. Accordingly, local knockdown of the mediator significantly alleviated disease symptoms [92]. Other extracellular mediators able to worsen RA progression as a consequence of P2X7R activation are cathepsins [93] and lysosomal proteases involved in collagen degradation and joint damage [94]. Other cell populations playing a pathogenic role in the bone and joint tissue that could be conditioned by P2X7R activity and impact RA progression include osteoblasts, osteoclasts, osteocytes, and human type B synoviocytes [95]. P2X7R activation in osteocytes results in apoptosis and bone remodeling, whereas in osteoclasts, it induces merging to form multi-nucleated and active cells. P2X7R stimulation in human type B synoviocytes mediates further triggering of inflammation and joint damage [95,96]. Inhibition of the receptor using different antagonists can ameliorate disease severity by locally reducing inflammation [95]. Finally, P2X7R activity enhances osteoblast function through a cell-autonomous mechanism linked to the production of lysophosphatidic acid (LPA) and cyclooxygenase (COX), which in turn stimulate osteogenesis and skeletal remodeling and might also contribute to articular damage in RA [97].

The relevance in RA of pathogenetic mechanisms involving P2X7R activity paved the way for clinical trials with pharmacological antagonists of the receptor. Despite potent inhibition of IL-1β release in patient-derived monocytes, phase II studies aimed at evaluating the clinical efficacy of orally administered P2X7R antagonists on signs and symptoms of RA were unsuccessful [98,99]. An important obstacle in the interpretation of these failures is the concomitant administration of a stable background of methotrexate or sulfasalazine together with the P2X7R antagonist. The combination with these immunosuppressants would probably not provide a cumulative effect, allowing for the establishment of a therapeutic benefit of P2X7R antagonism. Further studies will be needed to provide a more targeted clinical rationale in selected patients and establish the usefulness of P2X7R as a target in this complex disease.

### 3.3. Multiple Sclerosis (MS)

MS is a chronic inflammatory autoimmune-mediated disease of the central nervous system (CNS) characterized by extensive myelin destruction and oligodendrocyte loss ensuing in CNS lesions that lead to severe physical disability [100]. However, spontaneous myelin repair can occur, counteracting neurodegeneration and progressive disability [101]. Gain of function mutations in the *P2RX7* gene were associated with the development of the disease in a Spanish case-control study [71], while loss of function SNPs were hypothesized to confer protection [74]. Moreover, two non-synonymous SNPs coding variants of *P2RX7* determining a gain of function of the receptor were associated with significantly higher disease activity in a cohort of patients with relapsing-remitting MS [69] (Table 1). P2X7R is present on the surface of many cells within the nervous system, among them neurons, astrocytes, oligodendrocytes, Schwann cells, and microglia [102], thereby complicating the interpretation of data obtained in patients expressing functional variants of the receptor. Indeed, activation of purinergic signaling in astrocytes and microglia is one of the major drivers of neuroinflammation and myelin damage, which are necessary prerequisites in the development of MS [103]. In MS patients, P2X7R immunoreactivity is augmented in activated microglia/macrophages in the spinal cord, and eATP apparently contributes to the MS lesion-associated release of IL-1β from microglia/macrophages [104]. In secondary progressive MS, the analysis of frontal cortex tissues showed P2X7R downregulation in monocytes, which was interpreted as a mechanism to limit the long-lasting opening of the P2X7R channel and cell death, thereby enhancing pro-inflammatory signals and further damaging the CNS. Conversely, P2X7R was upregulated in astrocytes, resulting in enhanced secretion of the monocyte chemoattractant protein-1 (MCP-1), thus contributing to the maintenance of inflammatory mechanisms [102] (Figure 1). 

The most common animal model used to gain insights on MS pathogenesis is experimental autoimmune encephalitis (EAE). In this model, triggering the P2X7R pathway by ATP or BzATP administration led to inflammation, oligodendrocyte death, and neural damage, while the use of P2X7R antagonists led to decreased inflammation and demyelination and ameliorated the symptoms [105]. In addition, *P2rx7* gene deletion reduced the development of the diseases in mouse models, preventing inflammation and neural injuries [105,106]. However, opposite results have been obtained in another study showing how *P2rx7* deletion causes disease exacerbation due to loss of lymphocyte homeostasis [107]. The interaction of CD4^+^ T cells isolated from the spinal cord of EAE rats with astrocytes resulted in increased Ca^2+^ activity in astrocytes via P2X7R activation, a phenomenon likely contributing to neuropathology [108]. Using a cuprizone model of demyelination, P2X7R in M1 microglial cells and reactive astrocytes was shown to contribute to T cell-independent inflammation and demyelination. Nevertheless, P2X7R blockade during remyelination did not ameliorate myelin recovery nor attenuate glial reactions and inflammation in white matter [109]. In mice with EAE, P2X7R was recently shown to characterize gut-imprinted T helper cells with specific effector functions and targeting features in the CNS. P2X7R-expressing T cells selectively constituted white matter inflammatory foci in the brain stem, cerebellum, and spinal cord [110]. Whether P2X7R expression characterizes similar T cell infiltrates in the CNS of MS patients and might play a causative role in the immunopathological process needs further investigation. Altogether, these data point to a contribution of P2X7R activity in several cells and phenomena characterizing MS. It will be important to understand whether direct targeting of P2X7R or cell-specific P2X7R downstream events could be amenable to therapeutic intervention. 

### 3.4. Inflammatory Bowel Disease (IBD)

IBD is a disorder characterized by chronic intestinal inflammation and altered microbiota composition that is likely perpetuated by the onset of autoimmunity. The etiology is multi-factorial and could lead to the development of two different conditions, namely ulcerative colitis (UC) and Crohn’s disease (CD) [111,112]. One of the possible factors contributing to IBD onset is P2X7R, since it shapes the functional profile of several cell populations involved in the disease pathogenesis. The analysis of the P2X7R gain-of-function SNP His155Tyr and the loss-of-function SNPs Arg307Gln and Glu496Ala in a cohort of German subjects did not reveal significant associations with susceptibility to Crohn’s disease (CD) [113]. The receptor is widely expressed on the surface of immune cells populating the intestinal mucosa, including colonic mast cells as well as intestinal epithelial cells, thereby affecting enteric function at multiple levels [114,115,116]. ATP/P2X7R signaling is a central player in intestinal inflammation via inflammasome activation and the subsequent arousal of adaptive immunity [114]. *P2rx7* deletion or pharmacological blockade led to clinical amelioration of IBD, reduced inflammation, and prevented disease onset in different murine models of colitis induced by dextran sulfate sodium (DSS) or 2,4,6-trinitrobenzene sulfonic acid (TNBS) administration [117,118,119]. 

IBD is characterized by dysfunction of T cell subsets, and Treg cell impairment is a critical pathogenic element in intestinal inflammation. P2X7R activity limits the immunosuppressive potential of Treg cells and promotes conversion to Th17, which is a pivotal cell subset in intestinal immunopathology [46,120]. This mechanism was hypothesized to be important in contributing to IBD pathogenesis in a murine model in which IBD is triggered by the adoptive transfer of naïve CD4^+^ T cells into lymphopenic mice; the severity of the disease can in turn be modulated by the concomitant transfer of a variable number of Treg cells. In this experimental setting, *P2rx7*^−/−^ Treg cells proved to be superior controllers of disease induction compared to their wildtype counterparts, which readily converted to Th17 [46]. Consistent with the relevance of regulated P2X7R activity in Treg cells in intestinal homeostasis, purinergic antagonism or *P2rx7* deletion rescued Treg cell numbers and, on the other hand, increased tumor incidence in a mouse model of colitis-associated colorectal cancer (CA-CRC) [121]. The analysis of colonoscopy samples obtained from patients with CD showed increased P2X7R expression in the inflamed epithelium and lamina propria, where it colocalized with DCs and macrophages, suggesting P2X7R may represent a therapeutic target in CD [118]. Finally, the intestinal microbiota is an additional element that strongly impacts IBD susceptibility. In CA-CRC, a higher relative abundance of protective bacterial species was reported in *P2rx7*^−/−^ mice that was partially restored in wildtype mice by pharmacological antagonism of P2X7R, suggesting a possible secondary contribution of microbiota alterations dependent on P2X7R activity in CA-CRC development [122] (Figure 1).

### 3.5. Type 1 Diabetes (T1D)

Type 1 diabetes (T1D) is a chronic autoimmune disease that originates from the destruction of pancreatic β-cells, which are responsible for insulin production and therefore control glucose metabolism [123]. The immunological mechanisms underlying T1D development have been intensely investigated over many years with the aim of finding an effective alternative to replacement therapy with insulin or various transplantation approaches [124]. In this context, the eATP/P2X7R axis has emerged as an immunoregulatory circuit that might be exploited to promote organ-specific tolerance and mitigate T1D progression. In fact, pancreatic pathophysiology is profoundly influenced by purinergic signaling, and P2X7R is expressed in several pancreatic cell types [125]. In humans, P2X7R activity is involved in insulin and IL-1 receptor antagonist (IL-1Ra) secretion, thereby regulating β-cells function and protecting against IL-1β-induced cell damage, respectively [126]. Essentially, two preclinical models of T1D allowed for useful insights on the role of P2X7R: non-obese diabetic (NOD) mice, which are characterized by a genetic susceptibility leading to spontaneous autoimmune diabetes development [127]; and streptozotocin (STZ)-induced diabetes, in which the administration of the antibiotic streptozotocin induces β-cells destruction and diabetes onset [128].

One of the first pieces of evidence suggesting P2X7R involvement in T1D development was obtained in NOD mice, which were characterized by lower expression of the receptor and delayed disease onset [129]. Moreover, *P2rx7* deletion or Brilliant Blue G (BBG) administration hampered T1D development in mice treated with STZ; both conditions led to reduced hyperglycemia, islet destruction, and leukocyte infiltration in the parenchyma and pancreatic lymph nodes [130]. Fiorina and colleagues identified two loss of function *P2RX7* genetic variants as protective for T1D development (Table 1). The eATP/P2X7R axis was shown to be important in fueling the response of autoreactive CD8^+^ T cells, which are abundant in early-diagnosed T1D patients [70]. In pro-inflammatory or high glucose conditions, eATP was intensely released from Langerhans islets in vitro and in vivo during the early phases of diabetes development. P2X7R blockade with oxidized ATP reduced CD8^+^ T cell-mediated autoimmunity in vitro and delayed diabetes onset in NOD mice. It would be interesting to investigate whether pharmacological antagonism of P2X7R in T1D patients at the onset of disease might result in the generation of Treg cells capable of limiting inflammatory β-cells destruction [40,46]. Altogether, these data suggest that P2X7R is possibly involved in the recruitment of macrophages and DCs to the islets of Langerhans during the initial phases of the disease; these cells support the pro-inflammatory loop triggered by eATP production and migrate to the surrounding lymph nodes to mediate antigen presentation [131]. Ultimately, the adaptive immune system will take over, and cytotoxic T cells will mediate further β-cells destruction (Figure 1). Indeed, NLRP3 deficiency was shown to be protective against T1D development by hindering T cell activation and affecting the chemotactic migration of immune cells into the pancreatic islets [132].

### 3.6. Psoriasis

Psoriasis is a chronic inflammatory pathology primarily affecting the skin and characterized by epidermal hyperplasia and leukocyte infiltration [133]. The disease can be triggered by different adverse events, including stress, skin infections, and traumas, which mediate cell damage, resulting in the release of ATP in the extracellular environment and the subsequent triggering of the inflammatory loop necessary for the development of psoriatic plaques [134]. In psoriatic patients, P2X7R expression is increased in both nonlesional and lesional psoriatic skin compared with normal healthy tissue, and psoriatic responses could be initiated via P2X7R signaling. Moreover, treatment of skin migratory DCs with a P2X7R agonist potentiated the induction of Th17 cells [135]. Even though a recent study demonstrated that P2X7R was not essential for the development of imiquimod (IMQ)-induced psoriasis-like inflammation [136], P2X7R-dependent control of vascular endothelial growth factor (VEGF) and IL-6, both elements involved in the development of psoriatic lesions [137,138], suggests P2X7R activity could have implications for the pathogenesis and potential treatment of psoriasis (Figure 1).

### 3.7. Sjögren’s Syndrome (SS)

Primary SS (pSS) is a systemic autoimmune disease characterized by salivary and lacrimal gland dysfunction due to the infiltration of lymphocytes and other immune cells [139]. Autoantibodies targeting the Ro and La components of a ribonucleoprotein (RNP) complex are present in the majority of pSS patients and are associated with enhanced glandular inflammation, vasculitis, interstitial lung disease, and lymphoma [140]. A study on *P2RX7* functional polymorphisms and the Ro/La autoantibody response in pSS patients has recently found an interaction between a gain of function SNP/haplotype and seropositive pSS, whereas no associations were observed with loss of function SNPs/haplotypes (Table 1). The selective association of P2X7R gain of function with seropositive but not seronegative subjects suggests the pathogenetic function of P2X7R could relate to autoantigen exposure and inflammatory cytokine expression [75]. In peripheral blood mononuclear cells (PBMCs) from pSS patients, both mRNA and protein levels of P2X7R were found to be significantly higher compared to normal individuals, and pharmacological downregulation of P2X7R expression inhibited the release of proinflammatory cytokines by pSS PBMCs [141,142]. The expression of both P2X7R and inflammasome components was significantly higher in pSS gland specimens and was paralleled by increased expression of mature IL-18 in pSS saliva samples as compared to sicca syndrome non-SS and healthy subjects. The data correlated with anti-Ro/SSA positivity and lymphocytic sialadenitis focus score, establishing a link between P2X7R signaling and pSS exocrinopathy [142]. P2X7R is expressed in both acinar and ductal salivary cells, regulating the secretory function at different levels. P2X7R on the gland epithelium was hypothesized to contribute to SS pathogenesis as well [143]. In vivo administration of a P2X7R agonist induced apoptosis of salivary epithelial cells in wildtype but not *P2rx7*^−/−^, mice, suggesting P2X7R in the epithelial component could trigger an autoreactive antibody response (Figure 1). Accordingly, an attenuated immune cell infiltration was detected in *P2rx7*^−/−^ compared to wildtype mice perfused with the P2X7R agonist [144]. In primary mouse submandibular gland (SMG) epithelial cells, P2X7R activation induced NLRP3 inflammasome assembly and IL-1β release, a response that was absent in cells isolated from *P2rx7*^−/−^ mice, and in vivo P2X7R antagonism in a mouse model of salivary gland exocrinopathy ameliorated salivary gland inflammation and enhanced carbachol-induced saliva secretion [145]. These findings indicate that activation of the P2X7R contributes to salivary gland inflammation in vivo and further corroborate the possible relevance of the P2X7R as a target for the treatment of pSS.

## 4. Conclusions

In the last few years, the research on P2X7R has progressively expanded to encompass its function in different cell compartments, further establishing its essential role in many processes regulating immune system homeostasis and response. Pharmacological inhibitors able to block P2X7R downstream signaling could represent an effective therapeutic option for a broad spectrum of inflammatory and autoimmune diseases. Many in vivo studies highlighted how blockade or deletion of the receptor reduced inflammatory markers and alleviated disease manifestations [62,95,105,117,118,145]. Nonetheless, other studies have highlighted the possible detrimental effects of P2X7R inhibition in the complex pathogenesis of autoimmune disorders [53,81,106,107]. To date, clinical trials using P2X7R inhibitors have reached phase II for the treatment of RA [98,99] and CD [146]. However, further efforts will be needed to face challenges arising from the use of P2X7R inhibitors, among which are the possible toxicities and side effects and the establishment of the timing, duration, and dosage of the treatment.

## Figures and Tables

**Figure 1 ijms-24-14116-f001:**
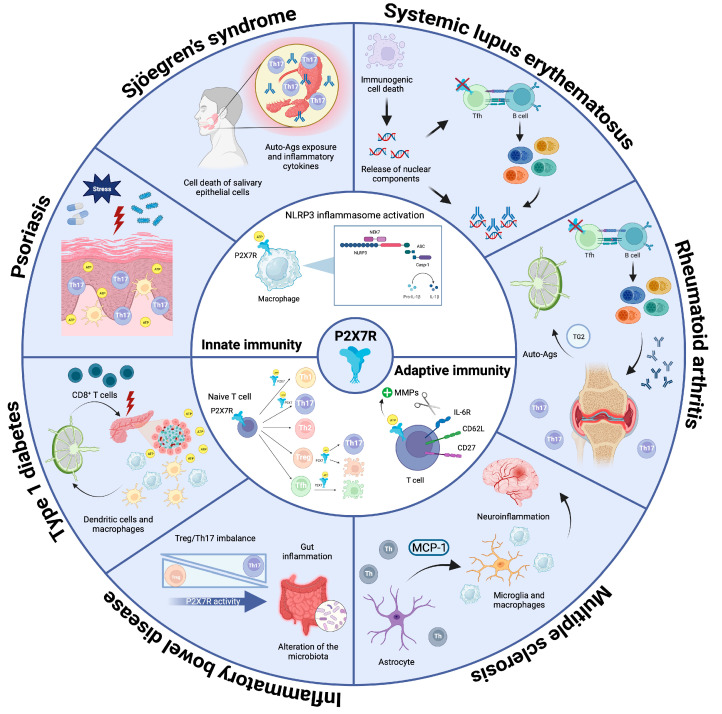
Function of P2X7R in autoimmunity. The diagram shows the general functions of P2X7R in innate and adaptive immunity (center) and peculiar pathogenetic mechanisms involving P2X7R activity in different autoimmune conditions. The possible involvement of P2X7R “loss of function” in Tfh cells in SLE and RA pathogenesis is indicated by a cross on P2X7R. Created with BioRender.

**Table 1 ijms-24-14116-t001:** Association of *P2RX7* SNPs with autoimmune diseases.

Polymorphism	Effect on Function	Implicated Conditions
rs1718119 A348T		SLE [67] RA [68] MS [69]
	Gain	
rs3752243 E496A	Loss	RA [68] Protective against T1D [70]
rs22390912 G464R	Gain	MS [69]
rs17525809 A76V	Gain	MS [71]
rs208294 H155Y	Gain	SLE [72] RA [73] MS [71]
rs7958311 R270H	Loss	Protective against T1D [70]
rs28360457 R307Q	Loss	MS [74]
rs2230912 A1405G	Gain	pSS [75]

## Data Availability

All data have been included.

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
