# Peer review of "The P2X7 Receptor in Autoimmunity"

_ijms, 2023, doi:10.3390/ijms241814116_

Round 1

Reviewer 1 Report

It is an interesting review, comprehensively yet concisely covering the main functions of P2RX7 in the innate and adaptive immunity and then discussing its interlocking with autoimmunity. The known aspects of P2RX7 involvement in specific autoimmune diseases are considered individually.  However, the commonalties (where they exist) could be presented more explicitly. Are there any congruences between the loss-of-function and gain-of function mutations across these different autoimmune diseases?

Given the significance of the functional tuning of P2RX7 in the context of autoimmunity, its known mechanisms should be considered, including post-translational regulation and MMP2-evoked cleavage, as these provide tight control on P2X7 activity.

The treatments aspects are considered, but mainly focussing on the experimental models, while the early clinical trials in RA, that were unsuccessful, should be discussed. Was the design of these trials based on the sufficient understanding of P2X7 receptor functions?

RE: the SLE subchapter

I am not convinced that the extensive description of SLE mouse models is relevant, given that only two of these models are being discussed in the following text. It detracts the reader from the main topic.

The manuscript contains a number of very long sentences. For example:

“Purinergic signaling is an evolutionary conserved mechanism involved in a variety 26 of physiological and pathophysiological processes; its functions have been intensely stud- 27 ied in the context of immunity and immunopathology given the widespread expression 28 of purinergic receptors in all types of immune cells. Purinergic receptors are divided in 29 two subtypes based on their binding affinity to different purine derivatives: P1 receptors 30 (A1, A2A, A2B and A3) are G protein-coupled receptors specific for adenosine, while P2 31 receptors are further divided in G protein-coupled metabotropic P2Y receptors (P2Y1, 2, 32 4, 6, 11-14) with binding affinities for ATP, ADP, UTP, UDP and UDP-glucose and P2X 33 ionotropic ligand-gated channels (P2X1-7) that bind ATP [1].”

These are just two sentences. Therefore, are hard to follow.

A few sentences would benefit from clarification. For example

(Lane 18) Autoimmune diseases are chronic disorders sharing inadequate organism self-tolerance by the adaptive immune system and tissue-destroying inflammation, thereby reducing life quality and expectancy.

I understand “sharing” relates to a common feature but this sentence would benefit from clarification.

(lane 49) “gene splice variants” should be rephrased, as splicing occurs at the transcript level.

(lane 364) “improved inflammation” could be rephrased.

Reviewer 2 Report

This review gives an extensive and updated scenario of the role of P2X7 receptor in autoimmunity

The review is well written and the reference list is appropriated 

No revision is required

Author Response

We thank the referee for appreciating our work.

Reviewer 3 Report

This is a comprehensive and detailed review of the role of purinergic P2X7 receptors in immunopathological processes with particular focus on autoimmune diseases such as systemic lupus erythematosus,  rheumatoid arthritis,  multiple sclerosis ,  inflammatory bowel disease ,  type 1 diabetes  and  psoriasis.

 I have only one minor comment:

The authors use two types of writing of gene for P2X7 receptor: „P2RX7“ in the first halve of the paper,  and „P2rx7“ in the second halve (since line 331), this should be unified.

Reviewer 4 Report

Grassi and Salina provided a very comprehensive and up-to-date review concerning P2X7 receptor involvemente in immunity, focusing on several autoimmune diseases.

I did not detect any particular issue and I only suggest to add a table to resume and schematize pathologies and the related P2X7 role.
